# Rehabilitation Outcomes for Patients with Motor Deficits after Initial and Repeat Brain Tumor Surgery

**DOI:** 10.3390/ijerph191710871

**Published:** 2022-08-31

**Authors:** Stanisław Krajewski, Jacek Furtak, Monika Zawadka-Kunikowska, Michał Kachelski, Marcin Birski, Marek Harat

**Affiliations:** 1Department of Physiotherapy, University of Bydgoszcz, Unii Lubelskiej 4, 85-059 Bydgoszcz, Poland; 2Department of Neurosurgery, 10th Military Research Hospital and Polyclinic, 85-681 Bydgoszcz, Poland; 3Department of Neurooncology and Radiosurgery, Franciszek Łukaszczyk Oncology Center, 85-796 Bydgoszcz, Poland; 4Department of Human Physiology, Ludwik Rydygier Collegium Medicum in Bydgoszcz, Nicolaus Copernicus University in Torun, Karłowicza 24, 85-092 Bydgoszcz, Poland; 5Department of Neurosurgery and Neurology, Ludwik Rydygier Collegium Medicum in Bydgoszcz, Nicolaus Copernicus University in Torun, M. Sklodowskiej-Curie 9, 85-094 Bydgoszcz, Poland

**Keywords:** brain tumor, repeat surgery, postoperative complications, function, rehabilitation

## Abstract

Repeat surgery is often required to treat brain tumor recurrences. Here, we compared the functional state and rehabilitation of patients undergoing initial and repeat surgery for brain tumors to establish their individual risks that might impact management. In total, 835 patients underwent operations, and 139 (16.6%) required rehabilitation during the inpatient stay. The Karnofsky performance status, Barthel index, and the modified Rankin scale were used to assess functional status, and the gait index was used to assess gait efficiency. Motor skills, postoperative complications, and length of hospital stay were recorded. Patients were classified into two groups: first surgery (*n* = 103) and repeat surgery (*n* = 30). Eighteen percent of patients required reoperations, and these patients required prolonged postoperative rehabilitation as often as those operated on for the first time. Rehabilitation was more often complicated in the repeat surgery group (*p* = 0.047), and the complications were more severe and persistent. Reoperated patients had significantly worse motor function and independence in activities of daily living before surgery and at discharge, but the deterioration after surgery affected patients in the first surgery group to a greater extent according to all metrics (*p* < 0.001). The length of hospital stay was similar in both groups. These results will be useful for tailoring postoperative rehabilitation during a hospital stay on the neurosurgical ward as well as planning discharge requirements after leaving the hospital.

## 1. Introduction

Primary central nervous system (CNS) tumors account for 1–2% of all cancers and 2–2.5% of cancer deaths. Approximately two-thirds of CNS tumors are benign (~50% meningiomas), and of the malignant tumors, ~40% are WHO grade IV glioblastomas, and ~40% are other types of glioma [1,2,3,4]. The treatment of these cancers has improved significantly over the last few decades, although the type of cancer and the patient’s age both influence outcomes. The five-year survival for patients with diagnosed malignant tumors increased from 23% to 36% between 1975–1977 and 2009–2015, with the largest gains for individuals aged 20–39 years, for whom five-year survival increased from 44% to 73%. However, five-year survival from glioblastoma has increased from just 4 to 7% compared with 32% for astrocytoma anaplastica, 54% for diffuse astrocytoma, and 88% for the most common non-malignant meningioma (malignant meningioma 66%) [1,5].

The current literature on the management of brain tumors highlights several knowledge gaps in the field. While there are very specific management algorithms for patients with newly diagnosed brain tumors, there is no uniform approach to the management of recurrences. For example, the Stupp protocol is now the standard of care for glioblastoma based on the histopathology and genetic profile of the tumor [6]. Nevertheless, despite the increasing efficacy of surgical and adjuvant treatment, these malignant tumors frequently recur. Many factors affect the risk and timing of recurrence: age, functional status, neurological symptoms, tumor size, growth rate and anatomical location, histopathological diagnosis, molecular profile, the extent of resection, and perioperative complications [1,2,3,4,7,8,9,10]. 

However, the management of recurrences is different from the management of newly diagnosed tumors. For example, anywhere between 25–59% of patients qualify for reoperation [7,11,12,13], highlighting the lack of specific guidelines for qualifying patients for repeat surgery. Furthermore, the value of repeat surgery is debated [14,15,16,17,18,19,20]. Frequently used criteria for evaluating the effectiveness of both surgical and adjuvant treatment are the assessment of performance and functional status. While several studies have evaluated the function and mental status of patients with primary brain tumors [8,21], few have evaluated these parameters in patients with recurrent tumors and after reoperation [22,23,24]. Of these, most data are on patients with glioblastomas, who often require early reoperation [6,7,11,12,25,26,27,28,29]. Likewise, there is a paucity of research on postoperative management and outcomes of patients after repeated brain tumor resections. 

It is therefore unknown how the operation affects functional state and motor skills and whether first or repeat surgery is associated with different postoperative rehabilitation outcomes. Therefore, here we addressed the lack of research in this area, taking into account the impact of postoperative complications on treatment outcomes. Recognizing and determining who is at risk of poor rehabilitation outcomes is important so that these patients can receive tailored goals and rehabilitation. Our primary aim was to compare the functional status, activities of daily living (ADL), motor skills, and gait efficiency of patients undergoing first brain tumor surgery and those undergoing reoperation. An additional aim was to assess the prevalence of complications affecting the rehabilitation course and time parameters such as the overall length of hospital stay (LOS), LOS after surgery, LOS in the intensive care unit (ICU), the time needed for rehabilitation, and the possible deterioration of basic motor skills after the first and repeated surgery.

## 2. Materials and Methods

### 2.1. Patient Cohort

The Bioethics Committee at the Military Medical Chamber approved the study protocol (No. 164/18). This was a single-center, prospective, observational controlled study (two intervention groups) with follow-up time from the day of admission to the clinic to the day of discharge. In total, 835 patients underwent operations for brain tumors in the 18 months between August 2018 and February 2020, 139 (16.6%) of whom required rehabilitation during their inpatient stay at the Neurosurgery Clinic. Three patients refused to participate in the study, and three died. The study, therefore, included 133 patients. In total, 103 patients (77.4%) underwent first tumor surgery (first surgery group), and 30 (22.6%) had reoperations (repeat surgery group). The inclusion criteria were patients who underwent the first or repeat brain tumor surgery, neurological deficits found, functional state worsened by surgery, and need for prolonged rehabilitation.

### 2.2. Patient Assessment

Primary variables included functional status and motor skills. Three scales were used to assess functional status: the Barthel index (BI), the Karnofsky performance status scale (KPS), and the modified Rankin scale (MRS). Each of these scales evaluates a different aspect of functional status. 

ADLs were assessed with the BI, which assesses self-reliance in eating, self-transferring (e.g., from bed to wheelchair), maintaining personal hygiene, using the toilet, washing, moving on flat surfaces and stairs, dressing, and controlling urine and bowel motions. Each activity is scored 5, 10, or 15 up to a total of 100 points: 0–20 means a severe condition, 20–80 indicates that the patient requires a varying degree of help, and >80 denotes an independent patient [30].

General condition and performance living with cancer were assessed with the KPS. The KPS is a 10-point graded scale, where 100 is full performance and 0 death. This scale assesses the impact of cancer on patient activity, taking into account their care and medical needs. A KPS score > 70 means that the patient can continue with normal activities and work with no special care needed; a KPS of 40–70 denotes that the patient is unable to work, can live at home, and can care for most personal needs, but that a varying degree of assistance is needed; and a KPS < 40 denotes that the patient is unable to care for self and requires the equivalent of institutional or hospital care. In these latter cases, the disease may be progressing rapidly [31].

The degree of dependence was assessed with the MRS. The MRS is a scale with 1-point increments, where 0 denotes no symptoms and 6 is death. This scale assesses to what extent and in what time dimension the patient requires care [32]. 

Gait efficiency was assessed with the 10-point gait index (GI). On the GI scale, 10 equals correct independent gait, and 1 equals impossible to achieve an upright vertical position; a score of 1–4 means the patient does not walk, 5–6 that the patient walks with the assistance of another person, 7–8 that the patient walks independently with orthopedic equipment, and 9–10 the patient walks on their own [21]. Patients were assessed with all four scales prior to surgery, immediately after surgery, and upon discharge. The first assessment usually took place on the day of admission; if the patient was on the ward for a long time before surgery, then the day before the procedure. Most evaluations of the impact of surgery on patient activity/performance compare patient condition prior to surgery and at discharge. We also assessed patients immediately after the procedure to determine the impact of the operation itself on the patient’s condition. Comparing performance status immediately after surgery and at discharge measures the effectiveness of rehabilitation and treatment of postoperative complications, including motor deficits, since the immediate postoperative period is the starting point for postoperative rehabilitation. The second assessment took place on the 2nd to 3rd day after the procedure, when the patient’s condition was no longer affected by the adverse effects of the anesthetic and immediate postoperative sequelae (vomiting, dizziness, headache, and short-term motor deficits). The third assessment took place on the day of discharge. The LOS of patients rehabilitated in the neurosurgery clinic ranged widely from 2 to 90 days. 

Motor skills—passive and active sitting and independent standing and gait—were assessed on an ongoing basis during rehabilitation. In this study, we evaluated individual motor abilities before surgery, a week after surgery, and at discharge. Given an estimated average duration of rehabilitation of about 14 days, evaluating function on the 7th day should capture the dynamics of improvement or deterioration. Secondary variables included the overall LOS, LOS after surgery, LOS in the ICU, the number of rehabilitation days, and the incidence of postoperative complications. The Landriel Ibañez classification was used to assess the severity and type of complications affecting the course of rehabilitation [33]. 

### 2.3. Statistical Analysis

All data are presented as mean ± SD or number (percentage) of participants. Normal distribution of the study variables was verified with the Shapiro–Wilk test. Differences in quantitative variables were determined with a nonparametric Mann–Whitney U-test. Relationships between categorical variables were determined with Pearson’s chi-squared test. To investigate group and time effects on functional activity, we used a two-way repeated measures analysis of variance (ANOVA) between groups (first and repeat surgery groups) and according to time (before surgery/after surgery/at discharge). Bonferroni’s test was used in the case of significant differences. A *p*-value < 0.05 was considered statistically significant. All calculations were carried out using Statistica 13.0 PL statistical package (StatSoft, Kraków, Poland).

## 3. Results

Of 835 patients with brain tumors, 139 (16.6%) required postoperative rehabilitation in the Neurosurgery Clinic, three died during their inpatient stay, and three refused to participate in the study. Finally, 133 patients were evaluated. The majority of patients (685, 82.0%) received the first surgery, and 103 of them (15.0%) required rehabilitation. One hundred and fifty (18.0%) patients required reoperation, 30 (20.0%) of whom needed rehabilitation. There was no significant difference in the proportion of patients requiring rehabilitation after first surgery and reoperation (*p* > 0.05).

The majority (89.5%) of patients had primary brain tumors, and 10.5% had metastases. In the group of patients undergoing surgery for the first time, non-malignant tumors were most common (64.1% of operations), especially meningiomas. Just over one-thirt of patients in this group had a diagnosed malignant tumor (including metastases). Of the 25.2% of operations for primary malignant tumors in the first surgery group, 61.5% were for WHO grade IV glioblastoma. Conversely, 60.0% of operations in the repeat surgery group were for malignant tumors (including metastases). The proportions of non-malignant and malignant tumors were significantly different between the first and repeat surgery groups (*p* = 0.018 and *p* = 0.009, respectively; Table 1).

There were no significant intergroup differences in gender, age, overall LOS, LOS after surgery, number of days in the ICU, and number of rehabilitation days. Five patients (20.0%) in the repeat surgery group and thirteen people (12.6%) in the first surgery group stayed in the ICU (Table 2).

Every fourth patient in the first surgery group and one in three in the repeat surgery group had postoperative complications. There was a difference between groups in the severity of complications according to the Landriel Ibañez classification (*p* = 0.047). Complications of surgery were the most common complications in both groups, being four times more common than medical complications in the first surgery group and 2.5-times more common in the repeat surgery group. The most common surgical complications were bleeding into the ventricular system (7), hydrocephalus (7), postoperative hematoma (6), cerebrospinal fluid leakage (6), and brain edema (4). Medical complications were a cardiorespiratory failure (6), dysphagia (PEG placement) (3), pulmonary embolism (1), and urinary tract infections (1). Temporary complications were more than twice as common as permanent ones in the first surgery group, while permanent complications were more common in the repeat surgery (*p* = 0.056). Paralysis and paresis were more common in the repeat surgery group both before surgery (*p* = 0.001) and at discharge (*p* = 0.009; Table 3).

Two-way repeated ANOVA revealed a difference in BI, KPS, MRS, and GI at different timepoints (before surgery, after surgery, and at discharge; all *p* < 0.001). After surgery, the first surgery group showed significantly lower BI, KPS, and GI scores and higher MRS scores compared with before surgery and at discharge. At discharge, the first surgery group was characterized by higher BI, KPS, and GI values and lower MRS values compared with after surgery (*p* < 0.001). There were no significant differences for BI, KPS, MRS, and GI before surgery and at discharge in the repeat surgery group. The GI (*p* = 0.014) was different between groups. The differences in MRS (*p* = 0.074) and BI (*p* = 0.079) scales were also close to statistical significance (Table 4 and Figure 1).

The mean BI score for the first surgery group was in the independent range (BI = 86.5), while the average BI score of patients in the repeat surgery group indicated a lack of independence (BI = 75.7) before surgery. After surgery, patients in both groups required a large degree of help (BI < 40) and required the help of various degrees at discharge (BI < 80; Table 4).

The mean KPS classified patients as able to carry on normal activities and work (KPS > 70) for both groups before surgery. The KPS decreased after surgery for both groups, who were classified as unable to work, able to live at home and care for most personal needs with a varying degree of assistance (KPS 40–70). The average KPS values were >70 for the first surgery group and <70 for the repeat surgery group at discharge (Table 4).

The mean MRS score classified the first surgery group as not significantly disabled and the repeat surgery group as slightly disabled before surgery. The first surgery and repeat surgery groups had average MRS scores of 3.5 and 3.7 after surgery, respectively, equating to moderate-severe disability. The mean MRS score of 2.3 in the first surgery group denoted slight disability, and 2.8 in the repeat surgery group had a moderate disability at discharge (Table 4).

The mean GI score classified patients as walking independently (GI 8.6) in the first surgery group and walking with orthopedic equipment (GI 7.0) in the repeat surgery group before surgery. Patients in both groups did not have the ability to walk (GI < 5) after surgery, and while first surgery patients could walk with orthopedic equipment (GI 7.2) at discharge, those in the repeat surgery group could walk with the assistance of another person for a distance of several dozen meters (GI 5.8; Table 4).

With respect to motor skills, the largest differences between groups were before surgery, especially with respect to standing and independent gait (*p* = 0.018 and *p* = 0.038, respectively). The percentages for individual activities decreased and equaled out in both groups after surgery, meaning that the worsening was more prevalent in first surgery patients. At discharge, all values in both groups improved compared with the week after surgery, but only passive sitting returned to pre-surgery values. Independent gait was possible for 70.0% of patients in the repeat surgery group and 80.6% of patients in the first surgery group before surgery compared with 50.0% of patients in the repeat surgery group and 60.2% of patients in the first surgery group at discharge (Table 5).

## 4. Discussion

There is vast literature on neurological and oncological rehabilitation but relatively little on the rehabilitation of patients with brain tumors [24,34], although several randomized controlled trials are now underway to address this knowledge gap [35,36,37]. The functional status and condition of patients with brain tumors are often compared with those occurring in patients after stroke [34,38,39,40,41,42,43,44,45] or craniocerebral injury [42,46,47]. Only a few studies have examined patients with tumor recurrence as a distinct group, usually evaluating reoperation outcomes with respect to overall survival and progression-free survival [48,49,50] or quality of life [22] and not usually in terms of rehabilitation outcomes. There has yet to be a comparative assessment of rehabilitation outcomes after initial and repeat surgery.

Here, we aimed to determine whether patients undergoing repeat surgery for a brain tumor are different, especially in the context of rehabilitation, from patients undergoing brain tumor surgery for the first time. The primary goal of our research was to compare the functional state of these two groups. We assessed independence, ADL, fitness in the context of the disease, and basic motor functions. An additional goal was to assess the incidence of complications and to determine time parameters: LOS, LOS after surgery, LOS in ICU, and duration of rehabilitation.

We found that patients undergoing reoperation were already in worse condition before surgery than patients undergoing surgery for the first time, especially with respect to the frequency of neurological deficits and gait disorders. These patients were also more likely to experience postoperative complications, especially the most severe ones (Landriel Ibañez grade III), which translated into differences at discharge over time. Although patients receiving first operations underwent a greater degree of functional deterioration, those requiring repeat surgery were in a worse functional state at discharge. However, these differences did not impact the LOS, which was similar for both groups. 

In total, 139 out of 835 patients receiving brain tumor surgery required a prolonged hospital stay and rehabilitation due to functional deterioration, similar to data reported previously from other centers [10,51,52,53,54]. With respect to most basic parameters, our group was similar to others presented in the literature [1,2,3,4]. Among patients receiving first operations, non-malignant tumors were the most common (64.1%), especially meningiomas. Most of the operated malignant tumors were gliomas, including the most common glioblastoma (61.5% of primary malignant tumors), which recur quickly and are, therefore, most likely to require reoperation. Half of the patients requiring reoperation underwent surgery for the recurrence of a malignant tumor. Malignant tumors are more likely to recur in men, and correspondingly, the gender distribution was almost equal in the first surgery group, while in the repeat surgery group, 60% were men and 40% were women.

Patients qualifying for reoperation were in a worse neurological condition before the procedure. Preoperative neurological deficits are an important determinant of ADL after surgery [55]. In the repeat surgery group, paralysis and paresis before surgery were present in as many as 70.0% of patients but in only half as many first surgery patients.

We used three commonly used assessment scales: the Barthel index (BI), the Karnofsky performance status scale (KPS), and the modified Rankin scale (MRS). We also assessed gait efficiency with the gait index (GI) [21] because gait re-education is often the main goal of postoperative rehabilitation. Prior to surgery, there were large intergroup differences in gait efficiency (according to the GI index) and the percentage of patients who could walk independently. The scores were significantly worse in the repeat surgery group. Gait efficiency impacted BI scores because, on this scale, three out of ten assessment items relate to movement. Repeat surgery patients were classified as requiring assistance, while first surgery patients were independent. Similar differences were observed when assessing the independence of patients using the MRS scale. Other authors have shown correlations between the BI and KPS [56] or MRS and KPS scales [57,58,59]. Using analysis of variance (ANOVA), we found that these scales assessed different aspects of function, and these correlations are not always obvious. Our groups did not significantly differ in KPS before the operation since the repeat surgery group had had time since their previous surgery to adapt to their physical state and function with their limitations.

After surgery, the percentage of patients with paresis increased in both groups. New motor deficits were reported in 36 patients, representing 4.3% of 835 operated tumors, but about 17% of all patients required a longer inpatient stay due to exacerbation of their neurological deficits (20.0% of patients undergoing repeat surgery and 15.0% after the first resection). 

Complications have been reported in anywhere between 9 and 40% of patients following brain tumor surgery [60], with a mode of around twenty percent. Data concerning the frequency of complications after brain tumor surgery are, therefore, quite variable and depend on the assessment criteria used and patient selection [33,54,61,62,63,64,65,66,67,68,69]. Cinotti et al. [70] showed that the preoperative functional state is a predictor of postoperative neurologic complications. We found a similar relationship. Postoperative complications were more common in the repeat surgery group. This difference was particularly notable for the most severe complications requiring reoperation and stay in the ICU, and, in 20% of patients requiring reoperation, the complications were permanent, compared with only 7.8% in the first surgery group. Severe and persistent complications occurred in 4.0% of all reoperated patients and 1.2% of patients undergoing initial operations. Immediately after the operation, functional status, as assessed by all scales, deteriorated significantly but also equalized between groups. 

At discharge, the condition of patients in both groups significantly improved relative to the scores after surgery but did not return to preoperative levels. Other authors have emphasized the effectiveness of postoperative rehabilitation as well as the need to continue rehabilitation in the post-hospital period [24,34,43,59,67,71,72,73,74]. Despite improvements, only half of the repeat surgery group were able to walk independently, as assessed by the GI scale, while in the first surgery group, 60.2% of participants were able to walk independently with orthopedic aids. There are some data suggesting that active sitting, standing, and independent gait have prognostic value and are strongly correlated with ADL [21]. Patients who can sit without support soon perform better in ADL [34,71], and standing greatly increases participation in ADL [43]. Independent gait not only determines participation in an active life but also provides a sense of self-confidence important for social and mental health. Speed of gait is a known survival factor in patients with brain tumors [72]. Average LOS values did not differ significantly between groups 18 days in the repeat surgery group and 16.5 days in the first surgery group, over three times longer than LOS after surgery in patients not requiring rehabilitation (5.1 days) [21]. 

Determining the risk resulting from the preoperative condition and whether the patient will undergo first or repeat surgery may be helpful in anticipating the resources needed for postoperative rehabilitation in neurosurgical wards. Given the possibility of an unfavorable course of postoperative treatment after brain tumor surgery, additional vigilance is required to identify and preemptively manage at-risk individuals. Our results highlight specific implications for physiotherapists and caregivers looking after brain tumor patients. When planning first or repeat brain tumor surgery, patients with motor deficits might need further rehabilitation in specialized centers or at home. Patients with new deficits after the first surgery are likely to find the postoperative period more challenging than those with such deficits on admission because disability is new for many of them. Our study is also useful for caregivers and social workers organizing further rehabilitation and appropriate facilities at home. Reoperated patients are at greater risk of postoperative complications, which should be taken into account in treatment and rehabilitation planning and inpatient resource distribution.

Our study has some limitations. First, the group sizes were very different (reflecting the incidence of recurrences and reoperations), so despite the clear differences between groups, there may be bias. In particular, the small numbers precluded subgroup analysis of patients with benign and malignant tumors, who may have experienced different outcomes. However, there are also some data showing no effect of tumor type on the course and results of postoperative rehabilitation [40], but this aspect would be interesting to study. Our study was single-center and, although covering 1.5 years and involving 835 patients, represents a relatively small number of patients since only 16.6% required postoperative rehabilitation. The tumor size and precise location were unknown, and these parameters would be interesting to study with respect to function. Finally, we did not explore some important neurological parameters, such as the degree of paresis before and after surgery, which was beyond the scope of this article.

## 5. Conclusions

Nearly one in five brain tumor resections required repeat surgery. Reoperated patients required prolonged postoperative rehabilitation as often as those operated on for the first time. The course of postoperative rehabilitation was more often complicated in the repeat surgery group, and the complications were more severe and persistent. These data must be taken into account in treatment and rehabilitation planning and inpatient resource distribution. Reoperated patients had the worst motor function and independence in ADL before surgery and at discharge, but deteriorations affected patients in the first surgery group to a greater extent for all metrics. Both LOS and rehabilitation time were similar between groups. For many patients with motor deficits, postoperative rehabilitation in a neurosurgery clinic is the first stage of improvement, but they might also need a referral for further rehabilitation in appropriate specialized facilities.

## Figures and Tables

**Figure 1 ijerph-19-10871-f001:**
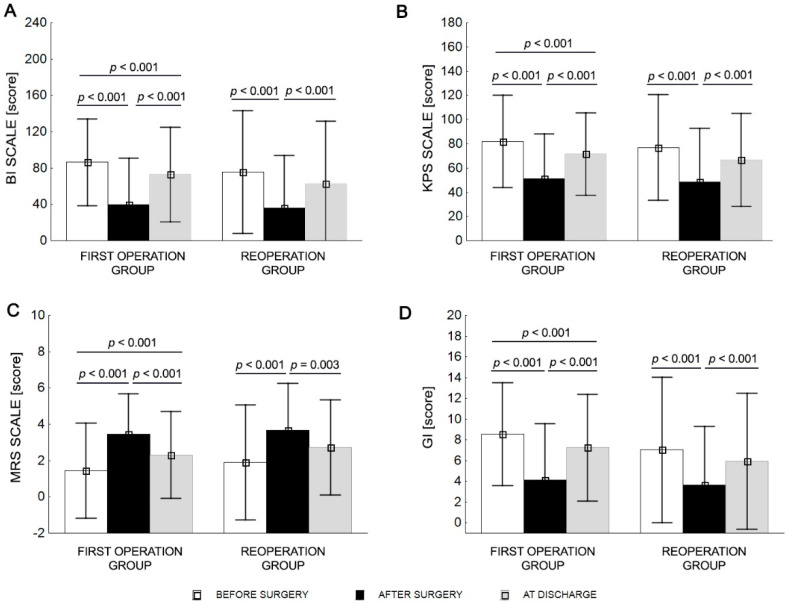
Mean values (±SD) before surgery, after surgery, and at discharge for BI (**A**), KPS (**B**), MRS (**C**), and GI (**D**).

**Table 1 ijerph-19-10871-t001:** Types of neoplasm in patients rehabilitated after first and repeat surgery for brain tumors (*n* = 133).

Type of Neoplasm	WHO Grade	First Surgery *n* = 103		Repeat Surgery *n* = 30		Total Operated Tumors *n* = 133		*p*-Value
		*n*	%	*n*	%	*n*	%	
**Benign tumors**								
Adenoma hypophysis				1	3.3	1	0.8	
Hemangioblastoma		5	4.9	1	3.3	6	4.5	
Hemangioma cavernosum		2	1.9			2	1.5	
Meningioma	I	21	20.4	2	6.7	23	17.3	
Schwannoma	I	15	14.6	1	3.3	16	12.0	
Other benign tumors		5	4.9			5	3.8	
**Low grade (WHO grade II)**								
Astrocytoma	II	3	2.9			3	2.3	
Diffuse astrocytoma	II	7	6.8	2	6.7	9	6.8	
Oligodendroglioma	II	1	1.0	2	6.7	3	2.3	
Ependymoma	II	3	2.9			3	2.3	
Meningioma atypicum	II	3	2.9	3	10.0	6	4.5	
Central neurocystoma	II	1	1.0			1	0.8	
**Total primary non-malignant tumors**		**66**	**64.1**	**12**	**40.0**	**78**	**58.6**	**0.018**
**Malignant tumors** **(WHO grade III, IV)**								
Anaplastic astrocytoma	III	5	4.9	3	10.0	8	6.0	
Anaplastic oligodendroglioma	III	1	1.0	3	10.0	4	3.0	
Anaplastic ependymoma	III	2	1.9	1	3.3	3	2.3	
Glioblastoma	IV	16	15.5	6	20.0	22	16.5	
Hemangiopericytoma	III			1	3.3	1	0.8	
Meningioma anaplasticum	III	1	1.0	1	3.3	2	1.5	
Supratentorial primitive neuroectodermal tumor	IV	1	1.0			1	0.8	
**Total primary malignant tumors**		**26**	**25.2**	**15**	**50.0**	**41**	**30.8**	**0.009**
Metastases *		11	10.7	3	10.0	14	10.5	0.915
**Total malignant tumors** **(primary + metastases)**		**37**	**35.9**	**18**	**60.0**	**55**	**41.4**	**0.018**
**Total operated tumors**		**103**	**100**	**30**	**100**	**133**	**100**	

* Metastases from lung cancer (6), breast cancer (3), intestinal cancer (1), ovarian cancer (1), and melanoma (2).

**Table 2 ijerph-19-10871-t002:** Demographic characteristics of the study participants and time parameters of treatment.

	First Surgery	Repeat Surgery	*p*-Value
Male *n* (%)	49 (47.6)	18 (60.0)	0.231
Female *n* (%)	54 (52.4)	12 (40.0)	
Age mean ± SD, [range]	50.0 ± 17.3 [19–83]	49.9 ± 12.3 [26–64]	0.763
Overall LOS (days)	20.1 ± 13.2 [4–92]	21.9 ± 16.6 [8–84]	0.925
LOS after surgery (days)	16.5 ± 12.7 [2–90]	18.0 ± 16.2 [5–79]	0.810
Days in ICU after surgery	0.7 ± 3.3 [0–31]	2.2 ± 7.7 [0–40]	0.494
Days of rehabilitation	12.5 ± 9.3 [1–53]	13.64 ± 12.2 [3–58]	0.777

Abbreviations: P, primary surgery; R, repeat surgery; LOS, the length of hospital stay; ICU; intensive care unit.

**Table 3 ijerph-19-10871-t003:** Complications (the Landriel Ibañez classification) and motor deficits.

	First Surgery		Repeat Surgery		All Operated Patients		*p*-Value First vs. Repeat
	*n*	%	*n*	%	*n*	%	
Patients with complications	26	25.2	10	33.3	36	27.1	0.387
Grade I	6	5.8	2	6.7	8	6.0	
Grade II	15	14.6	2	6.7	17	12.8	0.047
Grade III	5	4.9	6	20.0	11	8.3	
Surgical	21	20.4	7	23.4	28	21.1	0.732
Medical	5	4.9	3	10.0	8	6.0	0.302
Temporary	18	17.5	4	13.3	22	16.5	0.595
Permanent	8	7.8	6	20.0	14	10.5	0.056
Plegia/paresis							
Before surgery	37	35.9	21	70.0	58	43.6	0.001
At discharge	67	65.0	27	90.0	94	70.7	0.009

**Table 4 ijerph-19-10871-t004:** Activities of daily living, performance, self-reliance, and gait efficiency before surgery, after surgery, and at discharge.

Variable	Time	First Surgery	Repeat Surgery	Source	F	*p*-Value
		Mean ± SD	Mean ± SD			
BI	Before surgery	86.5 ± 23.9	75.7 ± 33.9	Group	3.1	0.079
	After surgery	39.4 ± 25.8	36.2 ± 28.7	Time	123.7	<0.001
	At discharge	73.0 ± 26.1	60.0 ± 36.2	G*T	1.1	0.324
KPS	Before surgery	81.8 ± 19.8	77.0 ± 21.8	Group	1.9	0.172
	After surgery	51.1 ± 18.5	48.3 ± 22.3	Time	99.8	<0.001
	At discharge	71.7 ± 17.0	66.2 ± 19.4	G*T	0.2	0.840
MRS	Before surgery	1.4 ± 1.3	1.9 ± 1.6	Group	3.2	0.074
	After surgery	3.5 ± 1.1	3.7 ± 1.3	Time	90.9	<0.001
	At discharge	2.3 ± 1.2	2.8 ± 1.3	G*T	0.5	0.631
GI	Before surgery	8.6 ± 2.5	7.0 ± 3.5	Group	6.2	0.014
	After surgery	4.1 ± 2.7	3.6 ± 2.8	Time	86.9	<0.001
	At discharge	7.2 ± 2.6	5.8 ± 3.3	G*T	1.6	0.211

Abbreviations: BI, Barthel index; KPS, Karnofsky performance status; MRS, modified Rankin scale; GI, gait index; SD, standard deviation; FS, first surgery; RS, repeat surgery; G*T, interaction effect Group*Time.

**Table 5 ijerph-19-10871-t005:** Functional state before surgery, a week after surgery, and at discharge (*n* = 133).

Motor Skills	First Surgery	Repeat Surgery	*p*-Value
	*n* (%)	*n* (%)	
Before surgery			
Passive sitting	103 (100%)	29 (96.7%)	0.066
Active sitting	102(99.0%)	28 (93.3%)	0.067
Standing	93 (90.3%)	22 (73.3%)	0.018
Independent gait	83 (80.6%)	21 (70.0%)	0.038
Week after surgery			
Passive sitting	93 (90.3%)	27 (90.0%)	0.482
Active sitting	82 (79.6%)	23 (76.7%)	0.386
Standing	62 (60.2%)	18 (60.0%)	0.494
Independent gait	40 (38.8%)	11 (36.7%)	0.449
At discharge			
Passive sitting	103 (100%)	29 (96.7%)	0.066
Active sitting	96 (93.2%)	26 (86.7%)	0.284
Standing	89 (86.4%)	21 (70.0%)	0.035
Independent gait	62 (60.2%)	15 (50.0%)	0.236

## Data Availability

All the data are presented within the manuscript.

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
