# Peer review of "Rehabilitation Outcomes for Patients with Motor Deficits after Initial and Repeat Brain Tumor Surgery"

_ijerph, 2022, doi:10.3390/ijerph191710871_

Round 1

Reviewer 1 Report

The paper entitled "Rehabilitation outcomes for patients with motor deficits after 2 initial and repeat brain tumor surgery", is an interesting systematic study about the functional state and course of rehabilitation of patients undergoing initial and repeat surgery for 17 brain tumors. The number of patients is adequate and the statistical data analysis is well done.

I think this paper is very useful in the evaluation of the disability of patients and I agree with the conclusions of the authors that write the need to undertake adjuvant treatment as soon as possible and the desire to extend the overall survival outside the hospital are factors modifying the time and goals of postoperative rehabilitation after brain tumor surgery.

The paper can be accepted for publication.

Reviewer 2 Report

Reviewer's comments:

The results of this manuscript are interesting. However, several major shortcomings are found. These questions are listed as follows:

1.    L.70  The authors intended to evaluate the complications of the patients who received rehabilitation after brain tumor surgery, not the factors affecting the rehabilitation course. Please delete the article's words "affecting the rehabilitation course" to avoid misunderstanding.

2.    L.70-72  The authors selected the secondary outcome variables to investigate the patients' clinical courses after initial or repeat brain surgery. In that case, the whole cohort of 835 patients (L.80) should be included, rather than only those who received rehabilitation after surgery. Please explain the reason for including only 139 patients.

3.    L.35-38  The sentences in this portion should be rewritten to improve the readability.

4.    L.57-58  The adjuvant treatment is out of the study's scope.

5.    L.98-99  Please explain the reasons for selecting these timing parameters and their clinical relationship.

6.   The authors should rewrite the result section to make it easier to read, e.g. the terms should be consistent with those in the tables (L.202), and the sentences must be completed (e.g. the timing? L.172-173 and 185-186).

7.    L.213-215  This manuscript critically lacks a discussion of the contribution of the study results for brain tumor patients.

8.    L.217-219  The authors did not mention the expected clinical significance of the selected outcome variables relating to treating brain tumors.

9.    L.231-236  There was a significant difference in the proportion of non-malignant and malignant tumors between the first and repeat surgery groups. Therefore, the study results may be biased because of these differences. Especially there were only 30 patients in the repeat surgery group. Thus, the authors should apply statistical methods to demonstrate that the significant relationship between the number of surgery and rehabilitation was not owing to the bias mentioned above.

10.   In the discussion section, especially between L.250-274, the authors repeated most of the description in the result section, lacking a detailed discussion of the study results, including related references. The author needs to rewrite this portion.

11.   L.288-297  Most of the description in this paragraph was not related to the study results, e.g. the discussion of tumor types and their prognosis were beyond the scope of this study (the relationship between the number of surgery and rehabilitation).

12.    In the discussion and the conclusion section, the authors are suggested to add the study results' importance and clinical implications to increase the manuscript's value.

13.   L.316-319  In the conclusion section, the description of adjuvant treatment's effects and the brain tumor's survival (L.316-319) should be deleted because these sentences were out of the study scope.

Reviewer 3 Report

The authors assessed functional state in terms of four parameters (BI, KPS, MRS and GI) before, after surgery and at discharge in both first surgery or reoperated groups of brain tumor patients, extending existing literature on presurgical functional performance of primary brain tumor patients. Both between group and across time of rehabilitation variable comparisons were conducted. Main findings of the article are rehabilitation time for both first surgery and reoperations are prolonged and equivalent, while complications, motor functions and independence in daily living were worse in reoperation group before surgery and at discharge. Deterioration after surgery affected patients in the first surgery group by all metrics. The manuscript is well written with great discussion interpreting results in the context of literature. The rationale/motivation and clinical implications of the study can be strengthened to elevate the study above mere descriptive statistics. Improvement suggestions are listed below:

Major issues:

1.       Please improve abstract and articulate the rationale/motivation for the study, before going into method.

2.       Please improve the introduction to focus on the rationale/motivation of the study and the specific knowledge gap the study intend to fill. For example, the authors stated “although when the disease is progressive or recurrent, there are currently no specific guidelines for qualifying patients for repeat surgery [7,14-19]. The value of repeat surgery is debated and not easy to assess due to difficulties in comparing patients undergoing one resection with patients undergoing multiple resections and controlling for confounding factors”. However the current study neither qualifies patients for repeat surgery or assesses the value of repeat surgery.

3.       Although it’s nice to compare the difference in terms of functional states in first surgery and reoperation groups and across time points in each group, it’s proably more clinically significant to know if patient demographic/clinical features and parameters of presurgical or post-surgical conditions can predict length of stay for rehabilitation, extent of functional improvement achievable at dischage and survival. Please consider adding a prediction model to your study with exisiting data.

4.       Please interpret the results as implications for clinical management and deicison making. For instance, what clinical implications are there for the result “both LOS and rehabilitation time were similar between groups”? Perhaps this would help with in-patient resource distribution, or perhaps this would help prepare the reoperated patients and their care-takers before surgery.

5.       The authors’s conclusion that “The need to undertake adjuvant treatment as soon as possible and the desire to extend the overall survival outside hospital are factors modifying the time and goals of postoperative rehabilitation after brain tumor surgery.” Cannot be supported by the data/results presented in the study.

Specific issues:

1.    Table 1. Are brain tumors metastases from lung/breast/intestinal/ovarian cancers or are those organs metastases from brain tumors?

2.    Please explain the difference and overlap between the three assessment scales (i.e. BI, KPS, MRS) and why they were selected for the study.
